# Are the Results of a Combined Behavioural and Surgical Treatment of Morbid Obesity Satisfactory and Predictable?

**DOI:** 10.3390/nu12071997

**Published:** 2020-07-05

**Authors:** Per G Farup

**Affiliations:** Department of Research, Innlandet Hospital Trust, N-2381 Brumunddal, Norway; per.farup@ntnu.no; Tel.: +47-948-18-603

**Keywords:** obesity, bariatric surgery, behavioural therapy, weight loss, biopsychosocial factors

## Abstract

Treatment of subjects with morbid obesity (Body Mass Index (BMI) > 40 kg/m^2^ or > 35 kg/m^2^ with obesity-related complications) often fails. This study explored the biopsychosocial predictors of dropout and weight loss during a combined behavioural and surgical weight-reduction program. Behavioural treatment for six months was followed by bariatric surgery and a visit six months after surgery. The success criterion was the loss of ≥50% of excess BMI above 25 kg/m^2^ (%EBMIL). Thirty-one men and 113 women with BMI 43.5 kg/m^2^ (SD 4.3) and 41.8 kg/m^2^ (SD 3.6), respectively, were included; 115 underwent bariatric surgery (Gastric sleeve: 23; Roux-en-Y gastric bypass: 92), and 98 had a follow-up visit six months after surgery. The mean %EBMIL at follow-up was 71.2% (SD 18.5). Treatment success was achieved in 86 subjects. Assuming success in 17 subjects who did not attend the follow-up visit (best possible outcome), 103 out of 144 subjects (72%) achieved successful weight reduction. Cohabitation was the only predictor of accomplishing surgery. Neither weight loss during behavioural therapy nor biopsychosocial factors were found to be clinically significant predictors of weight loss after surgery. The success rate of less than three in four subjects was unsatisfactory. There is a need to improve the regimen and to determine effective alternative interventions.

## 1. Introduction

Obesity is a worldwide disease with serious health consequences [1]. Subjects often lack motivation for treatment, and the treatment, being a behavioural intervention or bariatric surgery, often fails [2,3,4]. The weight loss following behavioural intervention is often suboptimal, without the anticipated health effects, and weight regain is a challenge [4,5]. Moreover, bariatric surgery is not always successful; 10–30% do not reach a relative percentage of excess Body Mass Index (BMI) loss (%EBMIL) of more than 50%, which has been mentioned as a goal for surgical treatment [6,7]. In general, obese subjects are hesitant to undergo lifestyle modification interventions, withdraw once the treatment starts, and back out from planned bariatric surgery. The predictors of treatment success have been studied to improve patient selection for treatment and develop interventions to improve compliance and outcomes. Thus far, the efforts have been disappointing [8].

In most countries, subjects with morbid obesity, especially those who are motivated to undergo bariatric surgery and are referred to specialised surgery units, accomplish combined treatment consisting of a behavioural intervention that aims to achieve a predefined weight loss, followed by bariatric surgery and a follow-up period of variable duration [9]. During the treatment period, some subjects drop out during behavioural treatment, some have insufficient weight loss, and some withdraw from surgery despite satisfactory weight loss. Reasons for dropouts and unsatisfactory results are poorly understood.

The aim of this study was to explore the biopsychosocial predictors of dropout from a combined behavioural and surgical weight reduction program and the predictors of weight loss six months after bariatric surgery.

## 2. Materials and Methods

The data used were a subset of data from in a study in subjects with morbid obesity, the MO-BiPS study (Morbid Obesity—Bio-Psycho-Social disorders) [10,11,12,13].

### 2.1. Study Design

The study had a prospective cohort design. The first part was six months with behavioural interventions aimed at weight loss, and motivation for and information about bariatric surgery. After finishing the first part, the participants were offered bariatric surgery and were followed for six months (the second part).

### 2.2. Participants and Inclusion Criteria

The study included subjects with morbid obesity (defined as BMI > 40 kg/m^2^ or > 35 kg/m^2^ with obesity-related complications) in the age range of 18–65 years referred from primary care to the specialised obesity unit at Innlandet Hospital Trust, Gøvik, Norway for the evaluation of bariatric surgery. Subjects who had undergone previous major gastrointestinal surgery or who had organic gastrointestinal disorders, problems with alcohol and drug abuse, major psychiatric disorders, or serious somatic disorders not related to obesity were excluded.

### 2.3. Interventions

At inclusion, the medical history was noted, a physical examination was performed, and a blood sample was taken. The participants filled in questionnaires about biopsychosocial disorders. During the first part of the study, i.e., the behavioural weight loss intervention, the participants had regular meetings, including both individual and group meetings, with nurses, doctors, nutritionists, and psychologists. They received dietary advice, physical activity programs, motivation for and information about bariatric surgery, and information about the health-related consequences of the procedure. During the last three weeks, the subjects followed a strict “crispbread diet” or a meal replacement powder diet containing 3765 kJ/day and unrestricted amounts of water, beverages without calories, and vegetables (except sweet corn, olives, and avocados), giving a maximum daily energy intake of 4200 kJ/day.

Bariatric surgery was performed as Roux-en-Y gastric bypass or Gastric sleeve [14,15]. A follow-up was conducted after six months.

### 2.4. Variables

At inclusion, the following biopsychosocial variables were collected:Demographic and anthropometric data: age (years), gender (male/female), height (m), body weight (kg), and body mass index (BMI; kg/m^2^);Weight loss was measured as reduction in BMI and relative loss of excess BMI above 25 km/m^2^ (%EBMIL). Treatment success was defined as %EBMIL ≥ 50%;Social factors: cohabitation (yes/no); education (number of years); employed (yes/no); smoking (daily/not daily);Dietary habits: The diet was assessed with a validated food frequency questionnaire, and the daily intake was based on the Norwegian food composition table [16]. Use of coffee was measured as number of cups per day. Alcohol intake was measured as more/less frequently than once a month;The physical activity score was calculated as the sum of two scores and classified as follows: easy activity (not sweaty/breathless)—none; <1 h/week, 1–2 h/week, and >3 h/week (scores 0–3); strenuous activity (sweaty/breathless)—none; <1 h/week, 1–2 h/week, and >3 h/week (scores 0, 3, 4, and 5). The total sum of the scores for physical activity was 0–8;Comorbidities: The participants were asked about the presence of 12 current or previous disorders (yes/no; scores: 0–12);Musculoskeletal pain from six parts of the body (scores 0–12);The World Health Organisation – Five Well-Being Index (WHO-5) (scores 0–100; scores ≤50 indicate low mood and scores ≤28 indicate likely depression) [17];Hopkins Symptom Checklist 10 to measure psychological distress (scores 1–4; scores ≥ 1.85 indicate mental distress) [18];Fatigue Severity Scale: A validated Norwegian translation of the Fatigue Severity Scale was used (scores 9–63; scores ≥ 36 indicate further evaluation) [19];Rosenberg Self-Esteem Scale: A validated Norwegian translation of the international questionnaire was used (scores 10–40; values <25 indicate low self-esteem) [20,21];Epworth Sleepiness Scale: A validated Norwegian translation was used (scores 0–24; normal 0–10; mild 11–14; moderate 15–18; severe 19–25) [22];Sense of Humour Questionnaire: A Norwegian version of the short form SHQ-6: “Attitudes toward humour” was used (scores 6–24) [23];Suter’s food tolerance questionnaire (scores 1–27; high score = good food tolerance) [24].

### 2.5. Statistics

The results are reported as mean (SD) and number (with proportion in percentage). Comparisons between groups were analysed with chi-square tests and *t*-test depending on the type of data. The predictors of weight loss were analysed with linear regression analyses. The analyses were performed with IBM SPSS Statistics for Windows, Version 25.0. IBM Corp: Armonk, NY, USA. *p*-values < 0.05 were judged as statistically significant.

### 2.6. Ethics

The study was approved by the Norwegian Regional Committees for Medical and Health Research Ethics, PB 1130, Blindern, 0318 Oslo, Norway (reference number 2012/966) and performed in accordance with the Declaration of Helsinki. All participants gave written informed consent before inclusion.

## 3. Results

### 3.1. Participants

Thirty-one men and 113 women with BMI values of 43.5 (4.3) kg/m^2^ and 41.8 (3.6) kg/m^2^ respectively (*p* = 0.024) were included in the analyses; 115 underwent bariatric surgery (Gastric sleeve 23; Roux-en-Y gastric bypass 92), and 98 had a follow-up six months after surgery. Figure 1 shows the classification of participants in the study.

### 3.2. Participants’ Characteristics at Inclusion

Table 1 gives the characteristics of the subjects at inclusion divided into those with and without bariatric surgery with comparisons between the groups. Twenty-nine subjects withdrew from bariatric surgery, of whom only three attended the hospital for the visit at the end of the behavioural treatment. Cohabitation was the only significant predictor of accomplishing bariatric surgery.

### 3.3. Weight Loss during the Two Treatment Periods

Table 2 gives the changes in BMI and %EBMIL during the study in 98 subjects with a follow-up 6 months after surgery. Eighty-six (88%) achieved a %EBMIL ≥ 50% from inclusion, which was defined as treatment success. Assuming that 17 subjects who were lost to follow-up after surgery had an %EBMIL ≥ 50% (best possible outcome), 103 out of 144 subjects included in the study (72%) achieved an %EBMIL ≥ 50%.

BMI at inclusion was positively associated with a reduction in BMI and negatively associated with %EBMIL. Table 3 gives the predictors of weight loss in detail.

### 3.4. Associations between Weight Loss after the Two Treatment Periods

Weight loss was significant after both treatment periods, but there were no significant associations between weight loss in the two periods. Table 4 gives the details.

## 4. Discussion

Several criteria for weight loss success after bariatric surgery have been used [3,7]. A commonly used success criterion of %EBMIL > 50% was used in this study; however, it is not considered ideal [7]. At best, only 72% of the subjects entering the study achieved a %EBMIL ≥ 50%. The proportion is probably lower since all subjects lost to follow-up after surgery were included as successfully treated. In most subjects, nearly all the weight loss was achieved six months after the surgery. However, since the maximum weight loss was observed 12 to 24 months after the surgery, a few more subjects could reach the treatment goal [25]. The results in this study are in accordance with larger studies with six month to ten year follow-up periods [2,3,25,26,27,28].

Looking separately at the two treatment periods, the weight loss during the first period was satisfactory with a mean %EBMIL of 19%. For all subjects included in the study, the weight loss was probably less because the weight loss in 29 subjects (20%) who dropped out could have been less than those who accomplished the regimen. Comparisons with other studies are difficult because the behavioural weight loss regimens vary in length and content and are not always completely described. In large, the weight reduction was as reported in other studies [25,27,28].

Short- and long-term weight loss after bariatric surgery has been thoroughly studied [2,3,4,25,26,27,28]. The results after the second treatment period are in accordance with studies with the same surgical procedures and observation time [25,26,27,28]. Comparisons are, however, difficult because not all studies specify if the preoperative weight loss is included in surgically induced weight loss.

Are the results satisfactory? More than one out of four subjects did not reach the anticipated weight loss target. Overall, the treatment was challenging, involving the stringent behavioural intervention, which was followed by a surgical procedure that demands drastic changes in dietary habits and lifestyle. Most subjects with morbid obesity lack motivation and therefore desist from such treatments, even if they are free of cost, as they are in Norway. Knowing that the subjects in this study represent a selected and highly motivated group given a treatment not accepted by all subjects with morbid obesity, the combined treatment is not optimal. The study demonstrates the challenges in the treatment of obesity. The combined procedure used in this study, which is probably the most effective therapy, is neither optimal nor accepted by all subjects with morbid obesity. The method needs improvement, and new treatment options that are more acceptable by the participants with less radical interventions are required and better results need to be achieved. However, notably, subjects with a significant weight reduction who withdraw from surgery and those without optimal weight loss after surgery achieve favourable health effects [4].

The high rates of dropouts and treatment failures call for predictors of failures. The dropout rate in this study during behavioural therapy (20%) was high but was as anticipated. A review reports dropout rates above 35% in half of the included studies [29]. The six month preoperative period with regular individual and group meetings with health personnel, weight registration, and reports about compliance was demanding. The subjects received in-depth information about not only the favourable health effects of the combined behavioural and surgical intervention but also the possible short- and long-term adverse events and the necessity of lifelong dietary and lifestyle changes. The solidarity and support in the groups might have increased compliance, whereas the information about possible negative consequences might have frightened away people. The only predictor of completing the behavioural treatment period and undergoing surgery was living with a cohabitant. Psychosocial support from the cohabitant might have motivated participation in the intervention. Psychosocial support has been shown to improve weight loss in trials with behavioural interventions [30].

Although the mean %EBMIL after the combined therapy was judged as satisfactory, a large variation was observed. Valid predictors of weight loss success could improve both the selection of subjects for surgery, the pre-treatment procedures, and per-operative interventions. In this study, BMI at inclusion was positively and negatively associated with the changes in BMI and %EBMIL, respectively [7,31]. The more obese an individual is, the greater the weight loss and the lower the loss of excess weight. No other clinically relevant associations were found.

There are numerous studies of various predictors of weight loss after bariatric surgery, such as personality, anxiety, depression, psychopathology, eating disorders, impulsivity, socioeconomic status, alcohol and drug abuse, cognitive function, cognitive treatment, comorbidity, metabolic diseases, and hormones [6,8,9,31,32,33,34,35]. Overall, the results are unsatisfactory; no clear predictors have been identified. This study supports the statement by Sarwer et al., “The relationship between preoperative psychosocial status and postoperative outcome is one of the most researched but least understood issues in the area of bariatric surgery” [8].

Preoperative weight loss has been a requisite for surgery at many obesity units. The purpose is to evaluate compliance with the necessary dietary changes before and after surgery, reduce complications and improve the outcomes [8]. The benefit from this mandatory weight loss before surgery is at best dubious [25,27,28]. In this study, neither the reduction in BMI nor the %EBMIL in the two treatment periods were significantly associated.

The strengths of the study were the inclusion of unselected, consecutive subjects at the only free of charge public obesity unit in the region, the use of a preoperative period with behavioural therapy according to the national standard and international recommendations, the use of validated questionnaires, the clear separation of the effects of the two treatment modalities, and the study of associations between the effects in the two treatment regimens. Details about the duration of obesity, the presence of comorbidities including psychopathology, the use of alcohol and drugs, biomarkers, complications, etc. were incomplete or unavailable. A longer follow-up period would have been desirable.

## 5. Conclusions

The overall success rate, defined as %EBMIL ≥ 50%, was less than 72%. Taking into account that the subjects referred to an obesity unit accepted a challenging weight loss treatment and were a selected and highly motivated group, the success rate was judged as unsatisfactory. No clinically useful predictors of success, including the outcomes of the pre-surgical behavioural intervention, were found. The results conform to other studies. However, further improvements are required in the regimen and other effective behavioural, psychiatric, medical, endoscopic and surgical interventions.

## Figures and Tables

**Figure 1 nutrients-12-01997-f001:**
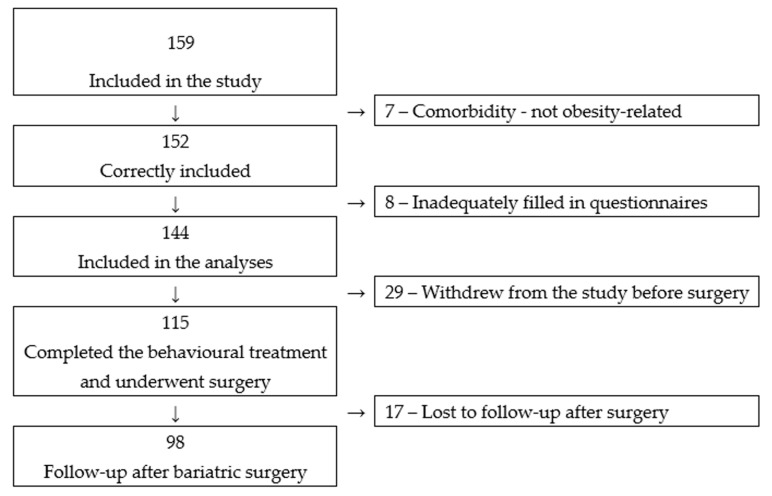
Participants in the study.

**Table 1 nutrients-12-01997-t001:** Characteristics at inclusion of subjects with and without bariatric surgery.

Characteristics	Bariatric Surgery		Statistics
	Yes (No 115)	No (No 29)	(*p*-Value)
Gender (men/women)	24 (21%)/91 (79%)	7 (24%)/22 (76%)	0.801
Age (years)	42.8 (8.7)	44.1 (9.0)	0.484
BMI at inclusion	41.9 (3.9)	43.0 (3.4)	0.202
BMI reduction after behavioural therapy ^1^	3.1 (1.7)	1.8 (1.8)	0.181
%EBMIL ^2^ after behavioural therapy (%) ^1^	19.3 (10.8)	8.7 (7.6)	0.094
Cohabitation (yes/no)	89 (77%)/26 (23%)	16 (55%)/13 (45%)	**0.021**
Education (years)	12.6 (3.6)	12.3 (4.3)	0.737
Employed (yes/no)	90 (78%)/25 (22%)	20 (71%)/8 (29%)	0.459
Coffee (cups/day)	3.1 (2.5)	2.4 (2.4)	0.216
Smoker (daily/not daily)	23 (20%)/92 (80%)	2 (7%)/27 (93%)	0.109
Alcohol ^3^	56 (49%)/59 (51%)	12 (41%)/17 (59%)	0.536
Physical activity (scores 0–8)	4.5 (2.2)	4.5 (2.2)	0.980
Burden of comorbidity (scores 0–12) ^4^	1.7 (1.3)	2.2 (1.6)	0.100
Musculoskeletal pain (scores 0–12)	4.4 (3.0)	3.8 (2.7)	0.338
WHO-5 (The World Health Organisation – Five) Well-Being Index (scores 0–100)	59 (18)	61 (16)	0.621
Hopkins symptom checklist 10 (scores 1–4)	1.56 (0.50)	1.68 (0.67)	0.296
Fatigue severity score (scores 1–7)	4.0 (1.6)	3.8 (1.7)	0.628
Rosenberg self-esteem scale (scores 0–30)	18.2 (5.1)	18.0 (5.7)	0.839
Epworth sleepiness scale (scores 0–24)	8.2 (4.7)	7.8 (4.8)	0.354
Sense of humour questionnaire (scores 6–24)	19.3 (2.3)	18.6 (3.8)	0.171
Food tolerance (Suter questionnaire) (scores 1–27)	24.1 (2.3)	23.8 (3.1)	0.643

Statistically significant *p*-values are written with a bold font. ^1^ The number of subjects in the groups with and without surgery were 110 and 3 respectively. ^2^ Excess Body Mass Index (BMI) reduction in percentage. ^3^ Alcohol more/less frequently than once a month. ^4^ The number of reported comorbidities.

**Table 2 nutrients-12-01997-t002:** Weight loss after the behavioural and surgical interventions in 98 subjects with a follow-up 6 months after surgery.

Treatment Period	BMI (kg/m^2^) Reduction	%EBMIL
1st period: Inclusion to Surgery	3.2 (1.7)	19.0 (10.9)
2nd period: Surgery to 6-month follow-up	8.6 (2.3)	52.1 (14.7)
Overall: Inclusion to 6-month follow-up	11.7 (2.7)	71.2 (18.5)

**Table 3 nutrients-12-01997-t003:** Predictors of weight loss after combined behavioural and surgical interventions. Linear regression analyses with BMI reduction and %EBMIL from inclusion to six months after surgery were used as dependent variables.

Independent Variables ^1^	Dependent Variables			
	BMI Reduction		%EBMIL	
	B (95% CI)	*p*-Value	B (95% CI)	*p*-Value
Gender (women/men)	0.21 (−1.15; 1.58)	0.756	−4.47 (−13.6; 4.7)	0.335
Age (years)	−0.02 (−0.09; 0.04)	0.479	0.42 (−0.02; 0.87)	0.062
Type of operation (sleeve/bypass)	0.62 (−0.80; 2.04)	0.389	−6.27 (−15.8; 3.3)	0.194
BMI at inclusion	0.28 (0.12; 0.43)	**0.001**	−2.29 (−3.30; −1.28)	**<0.001**
Cohabitation (yes)	−0.52 (−2.05; 1.01)	0.502	−3.70 (−13.96; 6.57)	0.477
Education (years)	−0.02 (−0.18; 0.13)	0.767	−0.27 (−1.30; 0.76)	0.602
Employed (yes)	0.94 (−0.34; 2.22)	0.148	−3.42 (−12.07; 5.23)	0.435
Coffee (cups/day)	0.02 (−0.22; 0.26)	0.849	−0.54 (−2.14; 1.06)	0.506
Smoker (daily)	0.18 (−1.26; 1.62)	0.809	1.37 (−8.31; 11.04)	0.780
Alcohol ^2^	0.152 (−0.97; 1.27)	0.788	−3.54 (−11.00; 3.93)	0.350
Physical activity (scores 0–8)	−0.11 (−0.36; 0.14)	0.385	−0.48 (−2.15; 1.19)	0.569
Burden of comorbidity (scores 0–12) ^3^	−0.28 (−0.65; 0.16)	0.228	0.11 (−2.63; 2.84)	0.940
Musculoskeletal pain (scores 0–12)	0.09 (−0.10; 0.27)	0.354	0.78 (−0.46; 2.03)	0.215
WHO-5 Well-Being Index (scores 0–100)	−0.03 (−0.06; 0.002)	0.064	−0.17 (−0.37; 0.03)	0.091
Hopkins symptom checklist 10 (scores 1–4)	0.70 (−0.39; 1.80)	0.205	4.17 (−3.2; 11.54)	0.264
Fatigue severity score (scores 1–7)	0.17 (−0.17; 0.50)	0.322	0.14 (−2.11; 2.38)	0.904
Rosenberg self-esteem scale (scale 0–30)	−0.01 (−0.12; 0.10	0.854	−0.19 (−0.94; 0.56)	0.620
Epworth sleepiness scale (scores 0–24	0.05 (−0.07; 0.17)	0.381	0.63 (−0.15; 1.41)	0.114
Sense of humour questionnaire (scores 6–24)	0.31 (0.07; 0.55)	**0.010**	0.32 (−1.32; 1.96)	0.701
Food tolerance (Suter questionnaire) (scores 1–27)	0.04 (−0.20; 0.28)	0.737	0.16 (−1.47; 1.76)	0.858

Statistically significant p-values are written with a bold font. ^1^ The variables were adjusted for age, gender, type of operation, and one by one of the other independent variables. ^2^ Alcohol consumption was measured as more/less frequent than once a month. ^3^ The number of reported disorders.

**Table 4 nutrients-12-01997-t004:** Linear regression analyses with BMI reduction and %EBMIL in the second treatment period as dependent variables.

Independent Variables ^1^	Dependent Variables			
	BMI Reduction 2nd Period		%EBMIL 2nd Period	
	B (95% CI)	*p*-Value	B (95% CI)	*p*-Value
Gender (women/men)	−0.13 (−1.27; 1.00)	0.815	−5.0 (−12.5; 2.4)	0.185
Age (years)	−0.07 (−0.12; −0.01)	**0.022**	−0.06 (−0.45; 0.33)	0.751
Type of operation (sleeve/bypass)	0.40 (−0.78; 1.58)	0.503	−4.1 (−11.9; 3.8)	0.305
BMI reduction 1st period	−0.03 (−0.32; 0.26)	0.840	---	
%EBMIL 1st period	---		0.07 (−0.24; 0.37)	0.664

Statistically significant *p*-values are written with a bold font. ^1^ All of the independent variables were included in the analyses.

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
