# Peer review of "Are the Results of a Combined Behavioural and Surgical Treatment of Morbid Obesity Satisfactory and Predictable?"

_nutrients, 2020, doi:10.3390/nu12071997_

Round 1

Reviewer 1 Report

1.in Materials and Methods: Please insert into this section, also as summary, data of cited informations lacking (row 4647). It was not helpful to look fo other papers to well and better undertand ll criteria of this manuscript.

2.In severl experiences one of the main useful predictor of success is the 'obesity age': how many time the patiens lived as obese? Do you ha ve this informations? please insert into the manuscript.

3.take a look in references. Some Author surname is not correct

Reviewer 2 Report

The author analyzed a six-months study in 145 morbid obese patients who received a combined behavioral and bariatric surgery weight-reduction program. There was a six-month follow up after the surgery which reported a 72% weight reduction success. The author rejected the original predictor, a biopsychosocial factor for the weight reduction or the behavioral therapy and concluded that cohabitation was the only predictor of successful surgery. The success rate of less than 75% was unsatisfactory and the author called for more effective alternative interventions.

The abstract was informative. The background clearly summarized the general information.

The method section was detailed and provided a concise description of the clinical study procedures. The methods followed the Norwegian Regional Committee for Medical and Health Research Ethics. The inclusion of ethics and the graphical description are a plus. Adequate information of a previous study that used the same experiment design was contained in the study supplements.

The statistical analysis and information about the software was sufficient.  

The results were thoroughly described, and the authors stated that both treatments reduced bodyweight significantly. During the periods between the two treatments, body weight was not reduced, and less than 75% of patients reached the goal weight reduction. The author did note that a variety of factors may explain results such as dropouts, length of behavioral treatment, some of the lost to follow-ups, etc. The data were convincingly presented.

The discussion and the conclusions are clear. The author rejected the original hypothesis that lack motivation caused the cessation of the treatment, since treatment is free in Norway. The lack of a financial motivation made the obesity treatment even more challenging. Since the treatment is known to be the most effective therapy of obesity in the present study, the author is calling for a very high criteria to define on effective treatment to obesity. Various potential predictors of the weight loss after bariatric surgery, such as personality, anxiety, depression, psychopathology, eating disorders, impulsivity, socioeconomic status, alcohol and drug abuse, cognitive function, cognitive treatment, comorbidity, metabolic diseases and hormones were examined with no clear predictors being identified. The reasons for the remaining study subjects are the least understood issue. Thus, the authors rejected the biopsychosocial factor as a reason for the lack of success they found no clinically useful predictors of success.

This is a nice study and analysis of the clinical treatment of morbid obesity, and the results conform to other studies. The author raised a serious question about the quality and a need for improving the regimen. Other effective behavioral, psychiatric, medical, endoscopic, and surgical interventions is highly demanding.

Suggestions:

The abstract may be reworded somewhat so that could be easier for readers to understand.
